# Injury Risk and Overall Well-Being During the Menstrual Cycle in Elite Adolescent Team Sports Athletes

**DOI:** 10.3390/healthcare13101154

**Published:** 2025-05-15

**Authors:** Azahara Fort-Vanmeerhaeghe, Montse Pujol-Marzo, Rai Milà, Berta Campos, Oriol Nevot-Casas, Pep Casadevall-Sayeras, Javier Peña

**Affiliations:** 1Faculty of Psychology, Education and Sport Sciences Blanquerna, Department of Sport Sciences, Universitat Ramon Llull, 08022 Barcelona, Spain; pepcs@blanquerna.url.edu; 2Segle XXI Female Basketball Team, Catalan Federation of Basketball, 08950 Esplugues de Llobregat, Spain; 3Faculty of Health Sciences Blanquerna, Department of Physiotherapy, Universitat Ramon Llull, 08022 Barcelona, Spain; raimonmv@blanquerna.url.edu (R.M.); bertacampos@uic.es (B.C.); oriolnc1@blanquerna.url.edu (O.N.-C.); 4Catalan Sports Council, Joaquim Blume Residence, 08950 Esplugues de Llobregat, Spain; 5Sport and Physical Activity Studies Centre (CEEAF), University of Vic–Central University of Catalonia, 08500 Vic, Spain; javier.pena@uvic.cat; 6Sport, Exercise, and Human Movement (SEaHM), University of Vic–Central University of Catalonia, 08500 Vic, Spain

**Keywords:** female, menstruation, young, adolescents, injury prevention

## Abstract

Background/Objectives: The impact of the menstrual cycle on the well-being and injury risk of young elite female athletes is poorly understood. This study assessed how the menstrual cycle phase influences perceived well-being and injury risk among young elite female team athletes aged 14–18 throughout a season. Methods: Wellness data, time-loss injuries, and menstrual cycle information were prospectively recorded for 59 young elite female team players throughout one season. The menstrual cycle was categorized into four phases using a standardized model: early follicular (menstruation), late follicular, early luteal, and late luteal (pre-menstrual) phases. Results: Significant differences were observed in wellness data, especially in sleep and fatigue, with poorer sleep quality and greater fatigue reported during the early luteal and late luteal (pre-menstrual) phases (*p* < 0.001). Furthermore, the luteal phase of the menstrual cycle was significantly associated with a higher incidence of sports injuries, particularly for joint/ligament and muscle/tendon injuries (*p* = 0.024 and *p* = 0.040, respectively). Conclusions: In elite female team athletes, poor sleep, increased fatigue, and elevated injury risk were significantly observed during the luteal phases of the menstrual cycle (early and pre-menstrual). These findings emphasize the importance of individualized monitoring and adaptive training strategies to mitigate the physiological effects of the menstrual cycle on athletic performance and injury risk.

## 1. Introduction

Women’s participation in sports has increased significantly in recent years, both at the amateur and professional levels [1]. Despite the growing participation of women in sports, research still disproportionately focuses on male athletes. Studies show that only 35% of the athletes examined are female [2]. One reason for this is the physiological variability caused by hormonal fluctuations from the menstrual cycle, often leading to fewer studies on women [3]. Additionally, when women are included, hormonal fluctuations are frequently overlooked, or testing is conducted during the early follicular phase, when both estrogen and progesterone levels are at their baseline lowest following menstruation, minimizing their effects [4]. Given the anatomical, physiological, and endocrine differences between men and women [5], specific research on female athletes is essential to individualize training and address current challenges in sports science, particularly concerning hormonal influences on performance and injury risk.

In recent years, there has been an increase in injury incidence among young athletes, especially in those under early sports specialization, which involves young athletes focusing intensely on one specific sport from an early age, often limiting or abandoning participation in other sports [6]. In many countries, specialized programs for athletes aged 14 to 18 both pre- and post-puberty exist. Participation in these specialized programs has been associated with a notable increase in injury incidence, especially among female athletes [7]. These increased injury rates may be partly attributed to key differences between men and women in sports, such as weight, height, and neuromuscular capabilities. Additionally, the menstrual cycle (MC) significantly influences female athletes’ training [8]. Fluctuations in hormones, such as estrogen and progesterone, throughout the cycle can affect various physiological parameters relevant to training. For example, changes in hormone levels can impact energy levels, muscle strength, ligament laxity, thermoregulation, and even mood, all of which can necessitate adjustments in training intensity, volume, and type at different phases of the cycle. Reducing injury incidence has become a critical part of training, particularly for young athletes, as injuries can reduce enthusiasm or induce fear of re-injury [9]. Therefore, early injury prevention strategies are essential to avoid long-term negative consequences [10].

Recently, a growing area of interest in athletic performance is understanding how the different phases of the MC may affect an athlete’s ability to perform. Many female athletes have reported noticing changes in their performance during both training and competition, especially in the late luteal and early follicular phases [11,12].

The current literature suggests that the menstrual cycle (MC) can influence both injury risk and well-being in female athletes [8,13,14]. The cyclic fluctuations of reproductive hormones such as estrogen and progesterone may affect musculoskeletal tissues like muscles, tendons, and ligaments [15,16]. For example, some studies have linked higher estrogen levels during the late follicular/pre-ovulatory phase to an increased risk of anterior cruciate ligament (ACL) injury due to potential reductions in connective tissue stiffness [17,18]. Conversely, other research indicates a higher incidence of ACL injuries during the early follicular/menstrual or late luteal phases [19,20]. Additionally, while Lago-Fuentes et al. [21] found that the follicular phase was associated with a greater likelihood of severe injuries among elite female futsal players, in contrast, Barlow et al. [22,23] observed that injury risk was significantly elevated during the luteal phase of the menstrual cycle among elite female professional footballers. Regarding well-being, some evidence suggests that athletes may experience higher levels of perceived exertion and fatigue during certain MC phases [13,14], highlighting the importance of tracking these subjective outcomes to understand athletes’ responses to training [8].

However, the current understanding of these relationships is limited. Most studies on this topic provide snapshots in time (cross-sectional), and few investigations specifically examine how athletes perceive their performance, menstrual symptoms, and factors like recovery and fatigue across the various phases of the MC. Therefore, there is a need for longitudinal research that tracks these subjective experiences in relation to objective measures and injury incidence throughout the menstrual cycle.

To our knowledge, no longitudinal study has examined the well-being and injury incidence in elite young female athletes in relation to their menstrual cycle. Therefore, this study aims to: (1) establish the characteristics of the menstrual cycle and (2) investigate how different menstrual cycle phases affect both injury risk and the perception of well-being in a group of young elite female athletes in the sports specialization stage.

## 2. Materials and Methods

### 2.1. Design

This study was conducted as a prospective cohort study (descriptive and longitudinal study) in which the menstrual cycle, injury incidence, and perceived well-being were monitored and recorded over the 2021-2022 season.

### 2.2. Participants

Fifty-nine young elite female athletes from a single high-performance sports center between June 2022 and June 2023 agreed to participate in the study. Participants were eligible for participation if they were female high-performance team sport players between 14 and 18 years old. However, several athletes who initially expressed interest could not be included or were subsequently removed from the study based on predefined exclusion criteria. Specifically, athletes were excluded if they had not yet begun their menstrual cycle (no menarche), were currently using combined hormonal contraception, or presented with a menstrual cycle-related pathology (irregularities). The criteria for menstrual irregularity were defined as: primary amenorrhea (no onset of menses by age 16), secondary amenorrhea (cessation of menstrual cycles for three or more consecutive months in the past year), or oligomenorrhea (menstrual cycles occurring at intervals longer than 35 days) [24]. During the course of the study, seven participants withdrew (dropouts) prior to the descriptive analysis. The reasons for these dropouts were: no menarche (n: 2), secondary amenorrhea (n: 2), using contraceptive pills (n: 2), and oligomenorrhea (1). The present study included 52 participants and 312 tracked menstrual cycles (six months). For data analyses (when comparing MC, injuries, and wellness), only cycles with a duration of 21 ± 7 days (between 21 and 35 days in cycle length) were included, as well as those with a bleeding phase duration between 2 and 7 days.

All participating athletes trained and studied at the same high-performance sports center, following similar daily training routines. These routines consisted of 8–10 sessions per week, each lasting 90–120 min, with 2–3 of these sessions dedicated to structured strength and conditioning. In addition to these sessions, the athletes played a weekend game, resulting in a total of approximately 16–20 h of combined practice and competition per week. Prior to the recruitment process, participants and their parents were provided with detailed information about the study, including the terms of their involvement. Informed consent was obtained from both parents or guardians and the participants before any data collection occurred. Participants were informed that they could withdraw their data up until the point of anonymization without facing any negative consequences. The Catalan Council of Sports granted ethical approval in July 2022.

### 2.3. Procedures

Six weeks before the beginning of formal data collection, the athletes underwent a familiarization period designed to ensure they understood how to track and report the three key study variables: menstrual cycle, injury incidence, and well-being perception. This familiarization process involved several components: Participants received detailed written instructions and visual aids, such as diagrams illustrating menstrual cycle phases and examples of how to complete the well-being scales. Secondly, a group meeting was conducted by the principal investigator and trained research assistants, providing a platform for interactive discussion, clarification of any doubts, and practical examples of data recording. Participants were encouraged to ask questions and practice using the recording tools (e.g., a dedicated section in their training logs or a secure online platform). The research team emphasized the importance of accurate and consistent reporting throughout the study.

### 2.4. Data Collection

This study utilized wellness, injury, and menstrual cycle data prospectively collected over a season (October to April).

#### 2.4.1. Menstrual Cycle Phase Identification

A method based on calendar counting was used to calculate the different phases of the menstrual cycle. The athlete recorded each month’s first day of menstruation and its duration through a mobile application (Clue Period Cycle and Tracker, Berlin BioWink GmbH). The principal investigator and three team members (MJ, MB, and ON) reviewed and recorded the required data weekly. The mobile tracking application employs an integrated algorithm that uses a standardized model to divide the menstrual cycle into four phases, relying on a presumed hormonal profile. In short, the total length of the menstrual cycle is used to calculate phases 2 and 3 retrospectively, based on prior research that assessed and predicted the duration of the follicular and luteal phases from thousands of menstrual cycles [22,23,25]. Menstruation is categorized as phase 1 (early follicular). The rest of the follicular phase is classified as phase 2 (late follicular), phase 3 covers most of the luteal phase (early luteal), and phase 4 corresponds to the pre-menstrual window (late luteal), defined as the five days leading up to the start of menstruation.

#### 2.4.2. Injury Incidence and Severity

Throughout the study season, all sports injuries sustained during matches and training sessions were recorded and monitored using the OSICS coding system [26]. Injuries were defined as any medical or physiotherapy consultation resulting from a match or training session [27]. An electronic version of the injury data registration form presented by Fuller et al. (2006) was used to record injury characteristics (severity, injury type, side, previous injury, recurrence level, cause, and circumstances of the injury) [27]. Injury severity was classified based on the number of days missed and was interpreted as follows: slight (0–1 days), minimal (2–3 days), mild (4–7 days), moderate (8–28 days), severe (>28 days). The medical staff completed an electronic injury form, and the principal investigator reviewed it weekly.

#### 2.4.3. Wellness Questionnaire

Each morning, before breakfast, the players recorded their level of well-being using the Hooper scale [28], which is based on the subjective analysis of the previous night’s sleep quality, stress level, fatigue level, and perceived musculoskeletal pain. Each question was rated individually with scores ranging from 1 (“Very, very low or good”) to 7 (“Very, very high or bad”). This questionnaire was sent to each player individually via a Google form.

### 2.5. Data Analysis

Statistical analysis was performed using the program SPSS for Windows (version 25.0, SPSS, Chicago, IL, USA). Menstrual cycle, injury incidence, and perceived well-being outcomes included all participants who completed the full follow-up period and the final assessment. Quantitative variables that followed a normal distribution were presented as mean, standard deviation (SD), and confidence interval (CI). For these variables, Kolmogorov–Smirnov was used to assess normality, along with Q-Q Plot normal distribution graphics. Frequencies and percentages described the type of injury. Linear one-way mixed models were used to analyze the influence of the menstrual cycle on perceived well-being outcomes. The model estimation method was restricted to maximum likelihood. To determine the relationship between the type of injury and the menstrual cycle, a Chi-squared analysis was carried out. All statistical tests of hypotheses were two-sided and employed a significance level of =0.05.

## 3. Results

### 3.1. Participants Description

Fifty-nine female athletes were studied throughout six menstrual cycles (approximately 6 months). The athletes practiced basketball (n = 23), volleyball (n = 21), and handball (n = 15), with a mean age of 15.9 (1.2) years and a BMI of 21.72 (2.06) kg/m^2^. Most participants did not use contraceptives (96.6%, n = 57) (Table 1). Regarding the menstrual cycle, the average duration of the menstruation phase was 4.56 (0.93) days, and the total cycle length was 28.13 (3.22) days, with no significant differences observed based on the sport practiced (*p* > 0.05).

### 3.2. Wellness Data

In analyzing the players’ wellness data, it is noteworthy that statistically significant differences were found in the dimensions of sleep and fatigue based on the menstrual cycle phase (see Table 2). Specifically, during the early luteal phase (3.07 ± 0.543) and the pre-menstrual phase (3.32 ± 0.518), sleep scores were significantly higher compared to the menstruation (2.24 ± 0.763) and late follicular (2.27 ± 0.784), with a *p*-value of less than 0.001. A similar trend was observed for fatigue, where scores in the early luteal phase (3.33 ± 0.718) and pre-menstrual phase (3.31 ± 0.682) were significantly higher than those during menstruation (2.79 ± 1.042) and the late follicular phase (2.80 ± 1.033), also with a *p*-value of less than 0.001. In contrast, no significant differences were observed in the dimensions related to stress and pain across the different phases of the menstrual cycle.

### 3.3. Injury Data

During the study period, a total of 126 injuries occurred (Table 3), with the majority classified as “slight” (n = 33, 26.4%). Regarding location, most injuries affected the lower extremities (n = 84, 65.6%). In terms of injury type, more than 85% were either joint (non-bone) and ligament injuries (n = 71, 55.5%) or muscle and tendon injuries (n = 42, 32.8%). Of the total injuries, 21% (n = 26) were re-injuries, with 61.9% occurring recently (early < 2 months). Regarding the cause of injury, 56.8% were traumatic, while 43.2% resulted from overuse. Concerning the injury mechanism, 64.3% (n = 81) occurred without any contact.

Significant differences were observed in injury severity (*p* = 0.029), overuse injuries (*p* < 0.001), and injury mechanism (*p* = 0.017) depending on the sport practiced.

Regarding the menstrual cycle, Table 4 presents the distribution of different sports injuries across the menstrual cycle phases. No significant differences were found between sports injuries and the menstrual cycle phase (*p* = 0.064). Joint and ligament injuries were the most common injury, with 70 reported cases. Most cases occur during the luteal phase, specifically in the early luteal (31 cases, 44.3%) and pre-menstrual (30 cases, 42.9%) phases. Conversely, these injuries are less frequent during the follicular phase, particularly in the menstrual (5.7%) and late follicular (7.1%) phases. Moreover, muscle and tendon injuries (the second most common) were observed across all phases, with the highest occurrence during the early luteal phase (40.5%). The pre-menstrual phase also showed a significant incidence (31.0%). In contrast, they were less frequent during the menstrual (11.9%) and late follicular (16.7%) phases.

Moreover, when comparing the follicular phase (phases 1 and 2) and the luteal phase (phases 3 and 4) in relation to sports injuries, the results were statistically significant (Figure 1). Specifically, a total of 99 injuries (78.4%) occurred during the luteal phase, while 27 (21.6%) occurred during the follicular phase (*p* = 0.012). In terms of injury type, 88.4% of joint and ligamentous injuries and 71.4% of muscle and tendon injuries occurred during the luteal phase, compared to 11.6% and 28.6%, respectively, during the follicular phase (*p* = 0.024 and *p* = 0.040, respectively).

## 4. Discussion

The present study examined the impact of the menstrual cycle on the well-being and sports injuries of 59 adolescent females practicing team sports. Significant differences were observed in wellness data, particularly in sleep and fatigue, with higher levels in the early luteal and pre-menstrual (late luteal) phases. In addition, a statistically significant association was found between the follicular and luteal phases of the menstrual cycle and the onset of sports injuries. Specifically, the luteal phase of the menstrual cycle was associated with a higher incidence of sports injuries, particularly in joint, ligament, muscle, and tendon injuries. These significant differences between the follicular and luteal phases highlight the potential impact of hormonal fluctuations on well-being and injury susceptibility.

### 4.1. Menstrual Cycle Characteristics in Adolescents

The scientific literature reports that the menstrual cycle length in adolescents can vary more than in adults, with a regular cycle length defined between 21 and 34 days [24]. Additionally, the length of menstrual bleeding is typically reported to range between 2 and 7 days. Therefore, the results of the present study, with an average menstruation duration of approximately 4.56 days and a cycle length of 28.13 days, fall within the normal ranges for adolescents.

### 4.2. Wellness and MC

Regarding well-being questionnaires, our findings indicate significant variations in sleep and fatigue across different menstrual cycle phases. Specifically, poorer sleep quality and greater fatigue were reported in the early and late luteal (pre-menstrual) phases compared to the follicular phases (early and late). These results align with previous research highlighting the menstrual cycle’s influence on sleep and fatigue in female athletes [29,30,31,32]. Hrozanova et al. (2021) found that female junior endurance athletes experienced poorer sleep quality during the late luteal phase than the follicular phase, supporting the notion that hormonal fluctuations impact sleep patterns [30]. Similarly, Romans et al. (2015) and Driver et al. (2008) reported that sleep disturbances, such as increased awakenings and reduced sleep efficiency, were more common in the luteal phase [31,33]. This could explain why sleep scores were significantly higher during the luteal phases in the present study. Regarding fatigue, our results are consistent with the findings of studies on elite athletes [29,31]. The pilot study on elite Australian football players also identified greater fatigue levels in the luteal phase [8], likely due to increased progesterone levels and their impact on sleeping energy metabolism. Moreover, Romans et al. (2015) suggested that fatigue during the luteal phase may be linked to disruptions in slow-wave sleep, the deepest and most restorative stage of non-REM (Rapid Eye Movement) sleep characterized by slow brain wave activity, leading to lower restorative sleep quality [31]. Taken together, these findings emphasize the importance of considering menstrual cycle phases in sleep and fatigue management strategies, particularly for female athletes. More tailored interventions, addressing common pre-menstrual symptoms that can disrupt sleep (e.g., mood, headache), alongside strategies to optimize sleep hygiene, could be explored to mitigate performance-related fatigue in the luteal phase. While individualized adjustments to training loads based on menstrual cycle phase hold promise, their feasibility and the degree of individual variability require careful consideration. Further research with larger sample sizes and objective sleep measures is necessary to deepen our understanding of these physiological variations and their implications for athletic performance. Moreover, hormonal fluctuations during the menstrual cycle can also have psychological and emotional impacts, such as changes in mood, anxiety, and stress levels [34,35]. Several factors may affect both perceived fatigue and the risk of injury. Future studies should explore the psychological effects of the menstrual cycle on female athletes, particularly how mood swings and stress relate to physical performance, training loads, and injury rates. Additionally, these studies should consider other sporting and personal life influences, such as congested competitive periods or busy exam schedules, which may also impact athletes’ well-being.

### 4.3. Injuries and MC

Concerning injury risk, the findings of this study indicate that no significant differences were initially found between sports injuries and menstrual cycle phases (*p* = 0.064). However, a specific analysis of the follicular and luteal phases revealed a significant association (*p* = 0.012). The majority of injuries (78.4%) occurred during the luteal phase, with joint and ligamentous injuries (88.4%) and muscle and tendon injuries (71.4%) being significantly more prevalent in this phase compared to the follicular phase (*p* = 0.024 and *p* = 0.040, respectively). These results align with the findings of Barlow et al. (2024) [36], who observed a higher injury risk during the luteal phase among elite female footballers. The increased incidence of injuries in this phase may be attributed to hormonal fluctuations, particularly the effects of progesterone, which has been associated with decreased neuromuscular control and altered ligament stiffness [4]. It is important to note that although no statistically significant association was found between injuries and the different phases of the menstrual cycle, we can highlight some key observations from Table 1. The shortest phase of the menstrual cycle, the pre-menstrual phase, lasts only 5 days and accounts for 42.9% (30 injuries) of total joint and ligament injuries and 30% (13 injuries) of total muscle injuries. These findings suggest that the pre-menstrual period is a high-risk injury span compared to other phases.

The literature presents conflicting evidence regarding the relationship between the menstrual cycle and injury risk. Some studies, such as those by Hewett (2007) and Balachandar (2017), have reported a higher risk of anterior cruciate ligament (ACL) injuries during the late follicular/pre-ovulatory phase when estrogen levels peak [17,18]. This has been attributed to estrogen’s role in reducing connective tissue stiffness and increasing ACL laxity. This has been attributed to estrogen’s role in reducing connective tissue stiffness and increasing ACL laxity. Similarly, Lago-Fuentes et al. (2021) found an association between the follicular phase and a higher likelihood of severe injuries among elite female futsal players [21]. Conversely, other studies have reported an increased incidence of ACL injuries during the early follicular/menstrual phase [19,20] or late luteal phase [20]. Martin et al. (2021) also examined injury incidence across the menstrual cycle in international female footballers [37]. Their findings revealed that muscle and tendon injuries were nearly twice as frequent during the late follicular phase compared to the early follicular or luteal phases. These discrepancies highlight the complexity of the relationship between hormonal fluctuations and injury risk, suggesting that neuromuscular function, fatigue, and training load may also influence susceptibility to injuries. Given these mixed findings, further research with larger sample sizes and controlled monitoring of hormonal levels is necessary to clarify the impact of the menstrual cycle on injury risk. Understanding these relationships could help optimize injury prevention strategies for female athletes, particularly by tailoring training and recovery protocols based on individual hormonal profiles.

Although this study found that joint, ligament, muscle, and tendon injuries were more prevalent in the luteal phase, it did not explore the specific mechanisms contributing to them. Detailed injury tracking and biomechanical analysis could help determine whether these injuries are due to altered biomechanics, reduced neuromuscular control, or fatigue. Identifying the specific mechanisms will be crucial for developing effective injury prevention strategies tailored to menstrual cycle phases.

### 4.4. Limitations, Strengths, and Future Research

When interpreting the results of this study, certain limitations should be considered. The perspective of this study is innovative and needed. Still, the results should be interpreted cautiously, as a larger sample from multiple teams or multicenter studies is needed for further conclusions. However, this study highlights the potential value of monitoring the menstrual cycle, paving the way for future research to enhance performance and reduce injuries in adolescent female team sport athletes. It emphasizes the importance of collecting continuous menstrual cycle data alongside standardized injury data. Current guidelines for recording injuries should include recommendations for incorporating menstrual cycle information.

Moreover, the homogeneous nature of the sample, all from the same high-performance center, is a notable strength. Still, it may also limit generalizability since the participants are only from three sports. Expanding the sample to include athletes from other disciplines (e.g., soccer, water polo) and varying performance levels (elite, sub-elite, recreational) could provide a more comprehensive understanding of how the menstrual cycle influences well-being and injury susceptibility across different contexts. Although this study was conducted over a single season and includes more injuries and person-days than many previous studies, the sample size is still relatively small, mainly when categorizing injuries into subtypes and examining their incidence, severity, and type across four phases. To replicate this study on a larger scale, it would be necessary to involve multiple teams or centers and multiple season observations.

Hormonal levels were not monitored in this research. It is impractical in a high-performance setting due to the challenges of invasiveness and the high costs of longitudinal data collection. Future research could explore the feasibility of wearable technologies that track menstrual cycle phases (using AI or hormone data) and provide personalized recommendations for training, sleep, and recovery based on individual hormonal profiles.

Finally, understanding the menstrual cycle as a vital sign of health in adolescent females is essential. It provides evidence-based indicators for normal puberty, menarche, cycle regularity, and bleeding patterns. Recognizing significant deviations from typical menstrual cycles is crucial, as these variations may signal underlying health concerns. In our study, some adolescent athletes experienced menstrual disorders, such as dysmenorrhea and irregular cycles, which could further impact their well-being and performance. Future research should explore the relationship between menstrual disorders and athletic performance, including the effects of heavy menstrual bleeding (menorrhagia) and painful periods (dysmenorrhea) on training capacity, fatigue, and injury risk. Notably, menstrual cycle education is crucial for elite adolescent athletes. Understanding their cycle is essential for optimizing performance, preventing injuries, and maintaining overall well-being, especially as young athletes experience significant physical and physiological changes. Educating them about this topic helps them recognize how hormonal fluctuations affect energy levels, recovery, strength, and injury susceptibility. This knowledge ultimately empowers them to make informed decisions regarding their training and competition.

## 5. Conclusions

This study is the first to prospectively examine wellness and injury rates across the four menstrual cycle phases in adolescent female athletes. Significant changes were observed during the luteal phase, including poorer sleep quality, increased fatigue, and a higher incidence of sports injuries. Given these findings, it may be beneficial to implement individualized monitoring and tailored training adaptations to support adolescent female athletes during this menstrual cycle phase. This could help optimize recovery strategies and injury prevention protocols. Future research should focus on specific interventions, such as sleep hygiene practices and workload modifications, to alleviate the effects of these physiological changes on athletic performance and injury risk.

## Figures and Tables

**Figure 1 healthcare-13-01154-f001:**
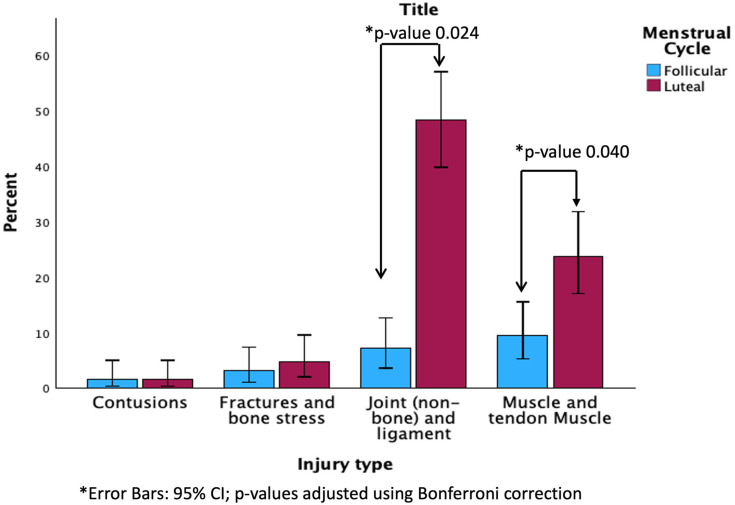
Follicular and luteal phases compared to sports injuries.

**Table 1 healthcare-13-01154-t001:** Participants characteristics.

	N	Mean (SD)
Age (years)	59	15.91 (1.23)
Years post PHV	59	3.76 (0.99)
Body mass (kg)	59	68.12 (9.23)
Height (m)	59	1.76 (0.08)
BMI (kg·m^−2^)	59	21.72 (2.06)
Training experience (years)	59	7.17 (2.12)
Menstruation days	59	4.56 (0.93)
Cycle length	59	28.12 (3.21)
Contraceptives (Yes)	2	3.30%
Basketball	23	38.90%
Handball	21	35.60%
Volleyball	15	25.50%

**Table 2 healthcare-13-01154-t002:** Players’ wellness data compared to menstrual cycle.

		N	Mean (SD)	CI 95%	*p*-Value *	Post Hoc Comparisons **	Effect Size (Eta-Squared)
Sleep	Phase 1. Menstruation	59	2.24 (0.763)	(2.02–2.45)	0.001	1 vs. 3.4; 2 vs. 3.4;	
	Phase 2. Late follicular	59	2.27 (0.784)	(2.04–2.49)			
	Phase 3. Early luteal	59	3.07 (0.543)	(2.89–3.24)			0.351 (0.237; 0.437)
	Phase 4. Pre-menstrual	59	3.32 (0.518)	(3.18–3.47)			
	Total	59	2.7 (0.825)	(2.58–2.82)			
Stress	Phase 1. Menstruation	59	2.53 (1.069)	(2.22–2.83)	0.755	Non-Significant	
	Phase 2. Late follicular	59	2.54 (1.057)	(2.24–2.84)			
	Phase 3. Early luteal	59	2.42 (1.022)	(2.02–2.81)			0.002 (0.00; 0.006)
	Phase 4. Pre-menstrual	59	2.5 (1.049)	(2.21–2.8)			
	Total	59	2.5 (1.086)	(2.35–2.66)			
Fatigue	Phase 1. Menstruation	59	2.79 (1.043)	(2.5–3.09)	0.001	1 vs. 3.4; 2 vs. 3.4;	
	Phase 2. Late follicular	59	2.8 (1.033)	(2.51–3.1)			
	Phase 3. Early luteal	59	3.33 (0.718)	(3.1–3.57)			0.80 (0.14; 0.251)
	Phase 4. Pre-menstrual	59	3.31 (0.682)	(3.11–3.5)			
	Total	59	3.04 (0.925)	(2.91–3.18)			
Pain	Phase 1. Menstruation	59	2.16 (0.925)	(1.9–2.43)	0.825	Non-Significant	
	Phase 2. Late follicular	59	2.19 (0.974)	(1.92–2.47)			
	Phase 3. Early luteal	59	2.2 (1.037)	(1.87–2.54)			0.001 (0.00; 0.005)
	Phase 4. Pre-menstrual	59	2.17 (0.915)	(1.91–2.43)			
	Total	59	2.18 (0.952)	(2.04–2.32)			

* *p*-value was calculated using F test; ** post hoc test Bonferroni.

**Table 3 healthcare-13-01154-t003:** Total number of injury incidence and severity.

		N	%
Severity	Slight (0–1 days)	33	26.40%
	Mild (4–7 days)	23	18.40%
	Minimal (2–3 days)	21	16.80%
	Moderate (8–28 days)	28	22.40%
	Severe (>28 days)	20	16.00%
Injury localization	Head and neck	6	4.69%
	Lower limbs	84	65.63%
	Trunk	10	7.81%
	Upper limbs	28	21.88%
Side	Both	15	12.30%
	Right	61	50.00%
	Left	46	37.70%
Injury type	Contusions	4	3.13%
	Fractures and bone stress	10	7.81%
	Joint (non-bone) and ligament	72	55.47%
	Muscle and tendon Muscle	42	32.81%
Previous injury	No	98	79.03%
	Si	26	20.97%
Recurrence level	Delayed recurrence (>12 months)	1	4.76%
	Early (<2 months)	13	61.90%
	Late recurrence (2–12 months)	7	33.33%
Cause	Overuse	51	43.22%
	Trauma	67	56.78%
Injury circumstances	Others	3	2.50%
	Practice	84	70.00%
	Match	33	27.50%
Injury mechanism	Ball contact	3	2.38%
	Object contact	5	3.97%
	Direct contact	18	14.29%
	Indirect contact	19	15.08%
	Non-contact	81	64.29%

**Table 4 healthcare-13-01154-t004:** Distribution of different sports injuries across the menstrual cycle phases.

	Contusions	Fractures and Bone Stress	Joint (Non-Bone) and Ligament	Muscle and Tendon Muscle	*p*-Value *
Phase 1. Menstruation	0 (0.0%)	3 (30.0%)	4 (5.7%)	5 (11.9%)	0.064
Phase 2. Late follicular	2 (50.0%)	1 (10.0%)	5 (7.1%)	7 (16.7%)	
Phase 3. Early luteal	2 (50.0%)	4 (40.0%)	31 (44.3%)	17 (40.5%)	
Phase 4. Pre-menstrual	0 (0.0%)	2 (20.0%)	30 (42.9%)	13 (31.0%)	
Total	4 (100%)	10 (100%)	70 (100%)	42 (100%)	

* *p*-value from Fisher’s exact test.

## Data Availability

The original contributions presented in this study are included in the article/supplementary material. Further inquiries can be directed to the corresponding author(s).

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
