# Peer review of "Injury Risk and Overall Well-Being During the Menstrual Cycle in Elite Adolescent Team Sports Athletes"

_healthcare, 2025, doi:10.3390/healthcare13101154_

Round 1
Reviewer 1 Report
Comments and Suggestions for Authors
I would like to thank the authors for writing an interesting research article that explores an important and current topic in sport. I offer line by line comments to help with a revision of the manuscript:
Line 38: 35% of what population? The way this is worded is not clear. Reword and make it clear what you mean.
Line 44: What current challenges?
Line 47: You need to explain what you mean by sport specialisation. I am not clear on what is meant, so it is safe to assume other readers will not know either.
Line 49: Increase in injuries in what sense? Is it that injuries increase when athletes join these programmes? Or has the reporting in the literature just increased? Not clear what is meant here.
Line 52: Does it significantly influence training? How so? This needs to be explained.
Line 60-67 needs to come after you reported the literature. So, first write about what we know in the context of menstrual cycle (phase) and injury and well-being (so a version of what you have in lines 68-85 at the moment, then you expose the gap (i.e. rewrite what is currently in line 60-67) and then you outline the aim of your study. That way you build a clear case for the need of your research.
Line 96: I would remove the word “calendar” here
Line 100: Change inclusion to participation
Line 103: Is there a citation you could provide that supports your decision to go with these criteria? Would be useful to know how you have come to define these criteria.
Line 99: I would start this section by saying how many participants actually took part. And then explained how many wanted to take part but had to be removed / dismissed for various reasons. And explain these reasons.
Line 114: Instead of using a negative here (you are currently saying that routines did not differ), it would be better to say that all participating athletes followed similar training routines that involved x y z (and then explain what you currently say about their routines).
Line 126-131: A few details are missing here. Please add: What do you mean when you say participants were familiarised six weeks before? What happened in the six weeks that followed? Who was involved in data collection that there needed to be training? What sort of training was this for the research team? What do you mean by positive feedback that was provided during the study?
Line 151: Where is this definition from? It would be useful to have reference provided here to support your decision to go with this definition.
Line 268-270: You say sleep scores were higher pre-menstrual, so I would be clearer here in your argument that sleep was less good. Otherwise it could be misunderstood as you claiming the opposite from what you write about later in the discussion.
Line 282: What is slow-wave sleep? This is mentioned as if the reader should know about this term, but the term not was introduced anywhere else in the manuscript.
Line 285: I think that of all the interventions you could be suggesting, this is quite a generic one that of course can be an additional consideration but that is not where I would see the real opportunity. If you have identified that sleep may suffer during the pre-menstrual phase, then you need to think about what sort of symptoms women might experience during that phase. Bloating, headache, etc. are all symptoms that have been reported in previous literature. Take a look at what you can find about this in the literature and then think about how interventions could support athletes. On the same point, I think it would be nice if training loads could be adjusted based on individual athletes’ cycles, but I wonder how realistic this is? And some athletes might not be affected at all, so then that would require for even more individualisation.
Author Response
I would like to thank the authors for writing an interesting research article that explores an important and current topic in sport. I offer line by line comments to help with a revision of the manuscript:
Thanks for your positive feedback on our manuscript and for taking the time to provide detailed comments. We appreciate your suggestions, and we have carefully addressed each point. We further enhanced the manuscript by refining the Introduction and Methods sections to provide greater clarity and detail.
We hope these improvements, in addition to the specific changes carried out, will strengthen the overall quality of the manuscript.
Line 38: 35% of what population? The way this is worded is not clear. Reword and make it clear what you mean.
Amended: “Despite the growing participation of women in sports, research still disproportionately focuses on male athletes. Studies show that only 35% of the athletes examined are female”
Line 44: What current challenges?
Added: “Given the anatomical, physiological, and endocrine differences between men and women [2], specific research on female athletes is essential to individualize training and address current challenges in sports science, particularly concerning hormonal influences on performance and injury risk.”
Line 47: You need to explain what you mean by sport specialisation. I am not clear on what is meant, so it is safe to assume other readers will not know either.
Added: “In recent years, there has been an increase in injury incidence among young athletes, especially in those under early sports specialization, which involves young athletes focusing intensely on one specific sport from an early age, often limiting or abandoning participation in other sports”
Line 49: Increase in injuries in what sense? Is it that injuries increase when athletes join these programmes? Or has the reporting in the literature just increased? Not clear what is meant here.
Changed: “In many countries, specialized programs for athletes aged 14 to 18, both pre- and post-puberty exist. Participation in these specialized programs has been associated with a notable increase in injury incidence, especially among female athletes”
Line 52: Does it significantly influence training? How so? This needs to be explained.
Added: “Fluctuations in hormones, such as estrogen and progesterone, throughout the cycle can affect various physiological parameters relevant to training. For example, changes in hormone levels can impact energy levels, muscle strength, ligament laxity, thermoregulation, and even mood, all of which can necessitate adjustments in training intensity, volume, and type at different phases of the cycle.”
Line 60-67 needs to come after you reported the literature. So, first write about what we know in the context of menstrual cycle (phase) and injury and well-being (so a version of what you have in lines 68-85 at the moment, then you expose the gap (i.e. rewrite what is currently in line 60-67) and then you outline the aim of your study. That way you build a clear case for the need of your research.
We have rewritten all the suggested lines.
Line 96: I would remove the word “calendar” here
Removed.
Line 100: Change inclusion to participation
Changed.
Line 103: Is there a citation you could provide that supports your decision to go with these criteria? Would be useful to know how you have come to define these criteria.
Reference added: Hillard P, 2014.
Line 99: I would start this section by saying how many participants actually took part. And then explained how many wanted to take part but had to be removed / dismissed for various reasons. And explain these reasons.
Changed.
Line 114: Instead of using a negative here (you are currently saying that routines did not differ), it would be better to say that all participating athletes followed similar training routines that involved x y z (and then explain what you currently say about their routines).
Changed.
Line 126-131: A few details are missing here. Please add: What do you mean when you say participants were familiarised six weeks before? What happened in the six weeks that followed? Who was involved in data collection that there needed to be training? What sort of training was this for the research team? What do you mean by positive feedback that was provided during the study?
Added: “Six weeks before the beginning of formal data collection, the athletes underwent a familiarization period designed to ensure they understood how to track and report the three key study variables: menstrual cycle, injury incidence, and well-being perception. This familiarization process involved several components: Participants received detailed written instructions and visual aids, such as diagrams illustrating menstrual cycle phases and examples of how to complete the well-being scales. Secondly, a group meeting was conducted by the principal investigator and trained research assistants, providing a platform for interactive discussion, clarification of any doubts, and practical examples of data recording. Participants were encouraged to ask questions and practice using the recording tools (e.g., a dedicated section in their training logs or a secure online platform). The research team emphasized the importance of accurate and consistent reporting throughout the study.”
Line 151: Where is this definition from? It would be useful to have reference provided here to support your decision to go with this definition.
Added: Fuller et al. (2006).
Line 268-270: You say sleep scores were higher pre-menstrual, so I would be clearer here in your argument that sleep was less good. Otherwise it could be misunderstood as you claiming the opposite from what you write about later in the discussion.
Thanks, it was a mistake. We have rephrased it: “Regarding well-being questionnaires, our findings indicate significant variations in sleep and fatigue across different menstrual cycle phases. Specifically, poorer sleep quality and greater fatigue were reported in the early and late luteal (premenstrual) phases compared to the follicular phases (early and late).”
Line 282: What is slow-wave sleep? This is mentioned as if the reader should know about this term, but the term not was introduced anywhere else in the manuscript.
Clarified: “Moreover, Romans et al. (2015) suggested that fatigue during the luteal phase may be linked to disruptions in slow-wave sleep, the deepest and most restorative stage of non-REM sleep characterized by slow brain wave activity, leading to lower restorative sleep quality [31].”
Line 285: I think that of all the interventions you could be suggesting, this is quite a generic one that of course can be an additional consideration but that is not where I would see the real opportunity. If you have identified that sleep may suffer during the pre-menstrual phase, then you need to think about what sort of symptoms women might experience during that phase. Bloating, headache, etc. are all symptoms that have been reported in previous literature. Take a look at what you can find about this in the literature and then think about how interventions could support athletes. On the same point, I think it would be nice if training loads could be adjusted based on individual athletes’ cycles, but I wonder how realistic this is? And some athletes might not be affected at all, so then that would require for even more individualisation.
Added: “More tailored interventions, addressing common pre-menstrual symptoms that can disrupt sleep (e.g., mood, headache), alongside strategies to optimize sleep hygiene, could be explored to mitigate performance-related fatigue in the luteal phase. While individualized adjustments to training loads based on menstrual cycle phase hold promise, their feasibility and the degree of individual variability require careful consideration.”
Reviewer 2 Report
Comments and Suggestions for Authors
This manuscript is potential but some improvements are necessary. While the menstrual cycle and injury risk in athletes are not new, the focus on adolescent athletes combined with a longitudinal design adds valuable novelty. The manuscript ‘s relevance is high, given the growing emphasis on the health and individualized training of young female athletes. The results show interesting trends, especially regarding wellness and injury incidence across menstrual phases, but the lack of effect size reporting and minimal discussion of practical implications weakens their impact. While the implications are valid, they are not fully explored in terms of specific training adaptations that could arise from the findings. The writing is clear but contains some redundancy.
Abstract: Suggestion: Add age range or specify "adolescents aged 14–18".
Line 24: “Significant differences were observed in wellness data…” → clarify "higher levels reported" (higher = worse?).
Keywords: consider adding some more keywords such as "Adolescents," "Injury prevention," "Sleep quality"
Line 38: “Only 35% of athletes studied” → awkward phrasing. Consider “Only 35% of sports science studies include female athletes” [2].
Line 41: “Low hormonal phases” → define this more scientifically.
Line 99: Clarify if the participants were randomized or recruited from a single center only.
Line 123-124: Ethical approval to be deleted (it is mentioned at Lines 402-404, but please add the day of approval).
Lines 136: Please justify using calendar method without other methods such as hormonal testing, BBT, ovulation kits, ultrasounds…
Line 152: “Injuries were was defined…” → Typo. Remove “was.”
Line 161: Hooper scale’s validity and reliability in adolescent populations?
Line 175: Needs more precision. Were covariates (e.g., age, sport type, training volume) controlled in the mixed model? If not, explain why.
Table 1: mark the right column with “%” for Contraceptives and "Sport".
Table 2: The p-values in Table 2 are most likely from the fixed effect of "Menstrual Phase" in a Linear Mixed Model, evaluated using a statistical test like a Wald test or F-test, but this is not well explained in the manuscript. Please clarify: Whether the models were random intercept only or included random slopes, what type of test used to calculate the p-values (likelihood ratio test? Wald test? F-test?), post-hoc pairwise comparisons were corrected for multiple testing (which they should be if they're comparing phases 1 vs 3 vs 4, etc.). Consider showing effect sizes, not just p-values. This would support practical relevance. Please add the significance level of p-value, specify the test used to determine p-value, specify the post-hoc… under the table.
Table 4: chi-square result (p = 0.064) is not statistically significant, yet authors emphasize trends too heavily. Some cells in Table 4 have very small expected counts, especially for contusions (n = 4 total). This violates assumptions of the Chi-square test, which requires expected counts >5 in each cell. In such cases, Fisher’s Exact Test or Monte Carlo simulation should be used instead.
It's not clear whether one test was run for all injury types, or separate tests for each. If multiple Chi-square tests were performed (e.g., for joint injuries, muscle injuries, etc.), correction for multiple comparisons (like Bonferroni) should be applied — but it wasn’t mentioned.
Later (lines 226–232), the authors group phases into follicular vs luteal and report new p-values (e.g., p = 0.024 and 0.040). This is another set of Chi-square tests, but again it's unclear whether correction for multiple tests was used.
Figure 1: Caption is weak, expand to explain findings.
Line 361: suggest adding a power analysis or rationale for sample size.
Author Response
This manuscript is potential but some improvements are necessary. While the menstrual cycle and injury risk in athletes are not new, the focus on adolescent athletes combined with a longitudinal design adds valuable novelty. The manuscript ‘s relevance is high, given the growing emphasis on the health and individualized training of young female athletes. The results show interesting trends, especially regarding wellness and injury incidence across menstrual phases, but the lack of effect size reporting and minimal discussion of practical implications weakens their impact. While the implications are valid, they are not fully explored in terms of specific training adaptations that could arise from the findings. The writing is clear but contains some redundancy.
Thank you very much for your insightful and constructive feedback on our manuscript. We have carefully considered all your suggestions and are committed to making the necessary improvements to strengthen the manuscript. We acknowledge your points regarding the lack of effect size reporting and the need for a more thorough discussion of practical implications. We calculated and included effect sizes in the revised manuscript and expanded the discussion to explore more specific training adaptations that could arise from our findings.
Provided that you identify any further areas for improvement or require any additional modifications, please do not hesitate to let us know.
Thank you once again for your time and valuable input.
Abstract: Suggestion: Add age range or specify "adolescents aged 14–18".
Added: “This study assessed how the menstrual cycle phase influences perceived well-being and injury risk among young elite female team athletes aged 14–18 throughout a season”.
Line 24: “Significant differences were observed in wellness data…” → clarify "higher levels reported" (higher = worse?).
Modified: “Significant differences were observed in wellness data, especially in sleep and fatigue, with poorer sleep quality and greater fatigue reported during the early luteal and late luteal (pre-menstrual) phases (p < 0.001).”
Keywords: consider adding some more keywords such as "Adolescents," "Injury prevention," "Sleep quality"
Thanks, added.
Line 38: “Only 35% of athletes studied” → awkward phrasing. Consider “Only 35% of sports science studies include female athletes” [2].
Amended: “Despite the growing participation of women in sports, research still disproportionately focuses on male athletes. Studies show that only 35% of the athletes examined are female. “
Line 41: “Low hormonal phases” → define this more scientifically.
Clarified:”Additionally, when women are included, hormonal fluctuations are frequently overlooked, or testing is conducted during the early follicular phase, when both estrogen and progesterone levels are at their baseline lowest following menstruation, minimizing their effects.”
Line 99: Clarify if the participants were randomized or recruited from a single center only.
Clarified: “Fifty-nine young elite female athletes from a single high-performance sports center between June 2022 and June 2023 accepted to participate in the study.”
Line 123-124: Ethical approval to be deleted (it is mentioned at Lines 402-404, but please add the day of approval).
Changed: “The Catalan Council of Sports granted ethical approval in July 2022.”
Lines 136: Please justify using calendar method without other methods such as hormonal testing, BBT, ovulation kits, ultrasounds…
While we acknowledge, as the reviewer pointed out, that the calendar method is not the most precise approach for determining menstrual cycle phases, it was employed in this study as a practical and non-invasive initial method for categorizing menstrual cycle phases in a large group of adolescent athletes in a real-world sports setting over a competitive season. This approach allowed for feasible data collection with minimal disruption to the athletes' training and competition schedules. Despite its limitations in precisely determining ovulation and cycle phase lengths compared to more direct measures, it can still reveal broader trends across estimated cycle phases in a group, providing valuable preliminary insights for future research employing more precise methodologies.
Line 152: “Injuries were was defined…” → Typo. Remove “was.”
Thanks, removed.
Line 161: Hooper scale’s validity and reliability in adolescent populations?
We acknowledge that the psychometric properties of the Hooper scale have been more extensively studied in adult athletes. However, a growing body of research suggests acceptable levels of reliability and validity for assessing subjective well-being in adolescent populations.
Line 175: Needs more precision. Were covariates (e.g., age, sport type, training volume) controlled in the mixed model? If not, explain why.
Linear one-way mixed models were used to analyze the influence of the menstrual cycle on perceived well-being outcomes, with menstrual cycle phase as the fixed effect and individual athlete as a random intercept to account for repeated measures. Covariates such as age, sport type, and training volume were not included in this initial model as the primary focus of this study is the within-subject variation in well-being and injuries across the menstrual cycle, a relationship poorly explored in adolescent athletes. We appreciate the reviewer's suggestion, and the potential influence of these covariates, which are more established in the literature, will be an essential consideration in future research.
Table 1: mark the right column with “%” for Contraceptives and "Sport".
Marked.
Table 2: The p-values in Table 2 are most likely from the fixed effect of "Menstrual Phase" in a Linear Mixed Model, evaluated using a statistical test like a Wald test or F-test, but this is not well explained in the manuscript. Please clarify: Whether the models were random intercept only or included random slopes, what type of test used to calculate the p-values (likelihood ratio test? Wald test? F-test?), post-hoc pairwise comparisons were corrected for multiple testing (which they should be if they're comparing phases 1 vs 3 vs 4, etc.). Consider showing effect sizes, not just p-values. This would support practical relevance. Please add the significance level of p-value, specify the test used to determine p-value, specify the post-hoc… under the table.
In accordance with your comments, we have added post hoc tests and effect sizes. We have also added a legend at the bottom of the table identifying the type of test used to calculate the p-value (F Test) and the multiple comparison tests using the Bonferroni test.
Table 4: chi-square result (p = 0.064) is not statistically significant, yet authors emphasize trends too heavily. Some cells in Table 4 have very small expected counts, especially for contusions (n = 4 total). This violates assumptions of the Chi-square test, which requires expected counts >5 in each cell. In such cases, Fisher’s Exact Test or Monte Carlo simulation should be used instead. It's not clear whether one test was run for all injury types, or separate tests for each. If multiple Chi-square tests were performed (e.g., for joint injuries, muscle injuries, etc.), correction for multiple comparisons (like Bonferroni) should be applied — but it wasn’t mentioned.
Thank you very much for your comment. Indeed, the p-value obtained refers to Fisher's exact test. The test performed was a comparison of proportions (Chi-square). No multiple comparisons were performed. We have added a legend at the bottom of the table to identify the test used.
Later (lines 226–232), the authors group phases into follicular vs luteal and report new p-values (e.g., p = 0.024 and 0.040). This is another set of Chi-square tests, but again it's unclear whether correction for multiple tests was used.
Thank you very much for your comment. In this case, multiple comparisons of proportions were performed, and the values were adjusted using the Bonferroni correction. We have added a legend at the bottom of the figure.
Figure 1: Caption is weak, expand to explain findings.
Caption expanded.
Line 361: suggest adding a power analysis or rationale for sample size.
It's a good observation. We'll take it into account for a future study. In this case, we were only able to work with the group of individuals we had available.
Reviewer 3 Report
Comments and Suggestions for Authors
We appreciate the authors' efforts to address a significant gap in sports medicine by prospectively examining the relationship between menstrual cycle (MC), wellness, and injury incidence in adolescent female athletes.
The results would be better by exploring whether MC phases differentially affect traumatic vs overuse injuries...looking more into hormonal impact on these.
Have you explored psychological impact during MC, using any validated measure to see its impact?
Please ensure consistent formatting in p-values and the tables.
Comments on the Quality of English LanguagePlease revise and proofread.
Author Response
We appreciate the authors' efforts to address a significant gap in sports medicine by prospectively examining the relationship between menstrual cycle (MC), wellness, and injury incidence in adolescent female athletes.
Thanks for your positive feedback and for recognizing the significance of our work in addressing the relationship between the menstrual cycle, wellness, and injury incidence in adolescent female athletes. We have carefully addressed all the points you raised and have made the suggested revisions to the manuscript. We hope that these changes have adequately improved the areas you identified. Should you find that further modifications are necessary, please do not hesitate to let us know. We would be happy to make any additional adjustments you deem appropriate. Thanks again for your time and valuable expertise in helping us enhance our work.
The results would be better by exploring whether MC phases differentially affect traumatic vs overuse injuries...looking more into hormonal impact on these.
Thanks for your insightful suggestion on differentiating between traumatic and overuse injuries. While this initial exploratory study focused on the broader relationship between the menstrual cycle and overall wellness/injury incidence in adolescent athletes, we recognize the importance of this distinction. We will prioritize this specific analysis in future research with larger sample sizes.
Have you explored psychological impact during MC, using any validated measure to see its impact?
Thanks for suggesting we explore the psychological impact during the menstrual cycle using validated measures. We agree this is important, but it was outside the scope of this initial study, which focused on physical wellness and injury. We appreciate you highlighting this valuable area and will strongly consider including psychological measures in future research.
Please ensure consistent formatting in p-values and the tables.
Thanks, changed.
Round 2
Reviewer 2 Report
Comments and Suggestions for Authors
The authors have adequately addressed all of my previous comments, and I believe the manuscript has improved significantly. However, I recommend that the authors carefully review the manuscript to ensure the following points are fully addressed: reformat all punctuation (dot to comma) and reference styles consistently; correct abbreviation usage and define each abbreviation only at its first appearance; acknowledge limitations more explicitly, particularly regarding hormonal phase estimation and potential confounding factors; streamline repetitive sections in the discussion; tighten the abstract and conclusion for greater impact; and improve the clarity of tables and figures. Additional minor comments are noted directly in the manuscript PDF.

Author Response
Many thanks to the reviewer for all the suggestions and detailed feedback, which have substantially improved the manuscript. We have corrected the issues with commas and periods; there was a formatting change that has now been rectified. Thank you again for your valuable input
Reviewer 3 Report
Comments and Suggestions for Authors
Thank you for addressing the comments.
Author Response
Thank you once again for all your valuable contributions.